# Female adolescent sexual reproductive health service utilization concerns: A qualitative enquiry in the Tema metropolis of Ghana

Innes Agbenu[1], Josephine Kyei[2], Florence Naab[1] *

**1** Department of Maternal and Child Health, School of Nursing and Midwifery, College of Health Sciences, University of Ghana, Accra, Ghana, **2** Department of Public Health, School of Nursing and Midwifery, College of Health Sciences, University of Ghana, Accra, Ghana

\* fnaab@ug.edu.gh

## Abstract

### Background

Evidence globally indicates that female adolescents face numerous sexual and reproductive health (SRH) risks. Utilization of sexual reproductive health services among adolescents is of global health importance and plays a crucial role in adolescent sexual reproductive health outcomes and their quality of life.

### Aim

The current study explored sexual reproductive health service utilization concerns among female adolescents in the Tema Metropolis in Southern Ghana using the Anderson and Newman Behavioural model of Health Service Utilization as a guiding framework.

### Methods

The study utilized a qualitative exploratory descriptive design. Purposive sampling was used to recruit female adolescents. In-depth face-to-face interviews were conducted using a semi-structured interview guide. In all, 12 interviews were conducted. Each interview lasted between 45 and 60 minutes. Interviews were audio-recorded, transcribed verbatim, and analyzed using thematic content analysis. Thematic analysis was guided by the constructs of the Anderson and Newman Behavioural model of health service utilization.

### Results

Utilization of sexual reproductive health services among female adolescents is low in the Tema metropolis. Factors such as unprotected non-consensual sexual activity or an unwanted pregnancy sometimes triggered the use of these services. Barriers to utilization identified include lack of awareness on sexual reproductive health services, unreliable sources of SRH information, underestimation of the severity of sexual reproductive health problems faced, unmet expectations and poor experiences with service providers.

**Funding:** The author (s) received no specific funding for this work.

**Competing interests:** The authors have declared that no competing interests exist.

## Conclusion

The current study identified poor utilization of sexual reproductive health services among female adolescents within the Tema metropolis of Ghana. There is the need to increase the number of adolescent health corners, increase awareness about SRH services among adolescents, improve parent-child SRH communication and provide adequate training for healthcare providers to improve provider attitude towards adolescent SRH service delivery in order to increase utilization of sexual reproductive health services among female adolescents in the Metropolis.

## Introduction

Sexual reproductive health services [SRHS] play crucial roles in adolescent sexual reproductive health outcomes and impacts the quality of life of adolescents. The World Health Organization [WHO] defines adolescence as a period of human development between ages 10 to 19 years [1]. This period of human development is said to be characterized by various risk taking behaviours such as sexual risk taking behaviours with their associated problems for most adolescents [2]. Globally, an estimated 1/6th of the world's population comprises adolescents. This represents 1.2 billion of the world's populace [3], with 88% of this estimate living in middle income countries and 16% residing in low income countries [3]. In sub-Saharan Africa, 33% of the population are within the adolescent group. In Ghana, adolescents constitute 21.9% of the population with the female ratio being slightly higher than that of the males [4].

Worldwide, adolescents face numerous health challenges including unintended pregnancies, abortions, HIV infections and other STIs [5]. Nearly 16 million girls between age 15 to 19 years and about 2 million girls below age 15years get pregnant each year [6]. Additionally, about 3. 9 million of these girls undergo unsafe abortions [7].

In Ghana, according to the multiple cluster indicator survey, 1 out of 10 adolescent girls become sexually active before age 15 [8]. The proportion of adolescent girls aged 15 to 19 years engaging in first sexual activity by age 15 has increased by 61.6% over the past 15 years [9]. A 2014 Ghana Health and Demographic survey reported that 14% of female adolescents aged 15 to 19 years had begun childbearing. Since 1980, a number of initiatives undertaken in Ghana to promote adolescent health led to the launching of several SRH programmes for adolescents including integrating adolescent health services into primary care, setting up of adolescent corners and distribution of adolescent health information leaflets by the Ministry of Health and the Ghana Health Service [9]. Despite these initiatives, adolescents in Ghana still face several challenges pertaining to sexual and reproductive health as a result of inability of the current strategies to provide adequate ASRH information and services [10]. Some adolescents resort to unsafe means of inducing abortion including inserting herbs into the vagina, drinking concoctions and boiled pawpaw leaves [10]. Also, adolescents have low risk perception for HIV leading to sexual risky behaviours [11]. Despite these sexual reproductive health risks, contraceptive use by adolescents in Ghana still remains low, with only 22% of sexually active adolescents using a contemporary contraceptive [12]. In the Tema Metropolis, the District Health Information Management Systems (DHIMS) report for the year 2018 indicated that there is low utilization of SRH services among adolescents within the metropolis. The report further indicated that although high attendance is recorded during educational visits to the various adolescent health centers, only few adolescents utilize the centers independently. In

addition, the metropolis recorded about 209 adolescent pregnancies and 31 new HIV diagnoses among female adolescents in the year 2018. In spite of all these, little has been done in Ghana on the utilization of sexual reproductive health services among female adolescents. Considering this knowledge gap, the study explored the utilization of sexual reproductive health services among female adolescents and related factors in the Tema Metropolis in Southern Ghana, using the Anderson and Newman Behavioural model of Health service utilization by [13] as a guiding framework.

The Anderson and Newman behavioural health services utilization model was used for the current study to explore factors that influence the utilization of sexual reproductive health services among female adolescents. The model suggests that a series of factors including predisposing factors, enabling factors and need factors influence the use of health services by people.

Female adolescents' use of SRH services first and foremost depend on predisposing factors which include demographic factors, social factors and health beliefs. According to the model, socio-demographic factors such as age, education, ethnicity, religion, occupation, family structure and marital status may influence the female adolescent either positively or negatively towards utilization of SRH services. The health beliefs held by these female adolescents also influence their use of SRH services.

Enabling fcators which include income, health insurance and availability of health related information is another factor which influence the utilization of health services. The ability of a female adolescent to utilize SRH services depends on the logistical aspects of obtaining care which include income, health insurance status, cost of services and whether health related information is readily available for them to make informed decisions.

The need factors aspect of the model refers to the functional ability of the female adolescent or existing health problems that will cause the female adolescent to utilize SRH services. According to the model, the need factor component is categorised into perceived need for health services and viewpoint of one's health. Perceived need for health services is whether the female adolescent thinks she needs to utilize SRH services for a particular health reason or based on her current functional ability. The viewpoint of one's health refers to how female adolescents think about their health status which will either make them appreciate the need to utilize SRH services or not.

## Materials and methods

### Research design

The study employed an exploratory descriptive qualitative design to explore the utilization of sexual reproductive health services among female adolescents aged 14 to 19 years in the Tema Metropolis. This method was used to elicit responses based on participant's experiences and social contexts and allowed participants to express their realities. It further explored and made meaning of the factors influencing the utilization of sexual reproductive health services among female adolescents.

### Study setting

The study was conducted in the Tema Metropolis in southern Ghana. The metropolis is an urban area known for high industrial activities and busy night life. The Tema metropolis is the second largest populated district in the Greater Accra region. The 2010 national population and housing census conducted in Ghana reported that the metropolis had 7.3% of the Greater Accra region's total population, with adolescents constituting 18.9 percent of the total population within the area. The Tema metropolis had only 4 adolescent centers located within public health facilities at the time of the current study and has been identified as a metropolis that

records high teenage pregnancies, illegal abortions and HIV infections among adolescents every year.

## Participants and recruitment

Twelve female adolescents aged 14 to 19 years, who could express themselves in English and received parental consent where needed were purposively sampled and included in the study. The purposive sampling technique was used because the researchers wanted to recruit female adolescents who had been living within the Tema metropolis for at least one year and are willing to take part in the study, as well as are able to receive parental consent to be part of the study where necessary. This is to ensure that participants with detailed information relevant to the study were recruited. The adolescent health officer in the Tema metropolitan health directorate served as a contact person who assisted with introducing female adolescents living within the metropolis to the researchers.

## Procedure for data collection

Data was collected using a semi-structured interview guide with open ended questions. The interview guide included the demographic characteristics of the participants as well as guiding questions to gather information on predisposing, enabling and need factors that influence utilization of SRH services among female adolescents. Interview guides were developed based on the constructs of the conceptual model used, reviewed literature and the objectives of the study. The interview guides were pre-tested among 3 female adolescents and further reviewed after the pre-tests. Interviews were conducted in English. In all, 12 interviews were conducted. Data saturation was reached after the 12th participant. Interviews were audio recorded after permission was sought from the participants. Field diaries were used to document major happenings, non-verbal cues demonstrated by participants and important incidents during the interviews. Interviews were audio recorded and at the end of each session, the interviews were played back to the participants to ensure that all important information had been gathered.

## Data analysis

Data analysis was done using thematic analysis. The analysis was used guided by the constructs of the Anderson and Newman Behavioural model of Health service utilization. This allowed for deep immersion into the data, making meaning of units and generating themes. Audio recordings were played repeatedly to get familiarised with the data and transcribed verbatim. The transcripts were read several times to identify patterns of similarity within the data. The identified common patterns were analysed in order to identify relationships and how they build up to support themes. Identified common patterns were compared with the original data to see if patterns and relationships identified were consistent. The identified common patterns and relationships were then condensed to build up sub themes and themes. In all, 3 themes and 12 subthemes were identified from the thematic analysis coducted. A final detailed report of the study results were written, highlighting the study findings, and supported with verbatim qoutes from participants.

## Ethical considerations

Ethical approval for the study was obtained from the Ghana health service ethics review committee (GHS-ERC049/11/19). Letters were sent to the Greater Accra Regional Health directorate to get permission for the study to be conducted in the region. Letters were also sent to the Tema metropolitan Health directorate to obtain permission for the study to be carried out

within the metropolis. Prior to data collection, the study objectives, information about the study and voluntary participation were explained to participants and parents. Participants were informed that their personal identifiers will not be included in the data to ensure anonimity. Participants were informed that they can withdraw from the study at anytime if they wish to. Written informed consent was obtained from participants and parental consent was also obtained from parents of participants where needed. Interviews were conducted between January and April 2020, on agreed dates as specified by the participants at their convenience.

## Findings

Three themes were identified in the study. These themes are predisposing factors, enabling factors and need factors that influence the utilization of sexual reproductive health services among female adolescents. Predisposing factors refer to factors that influence either positively or negatively utilization of services. Enabling factors are factors that make utilization of services easier. Need factors are situations that propel an individual to service utilization. Twelve subthemes identified were demographic factors, social factors, health beliefs, knowledge on sexual reproductive health, source of sexual reproductive health information and sexual reproductive health practices, affordability of services, health insurance and availability of health related information, perceived need for health services and view of one's health influencing female adolescent's utilization of SRH services within the metropolis. These are reported in Table 1.

### Demographic characteristics of respondents

Twelve participants between age 14 to 19 years took part in the study. Ten were Christians and two were Muslims. Nine were senior high school students and three were junior high school graduates. None of the participants was married at the time of data collection. All participants were living with either parents or guardians.

**Theme 1: Predisposing factors influencing the utilization of sexual reproductive health services among female adolescents.** Predisposing factors described factors that lead female adolescents to use sexual reproductive health services. Six categories of factors were described, namely social factors, health beliefs, knowledge on sexual reproductive health, source of sexual reproductive health information and sexual reproductive health practices.

**Table 1. Thematic structure.**

| Themes | Sub-themes |
|---|---|
| 1. Predisposing factors influencing the utilization of sexual reproductive health services among female adolescents | a. Social factors<br>b. Health beliefs<br>c. Knowledge on sexual reproductive health<br>d. Source of sexual reproductive health information<br>e. Sexual reproductive health practices |
| 2. Enabling factors influencing the utilization of sexual reproductive health services among female adolescents | a. Knowledge of adolescent sexual reproductive health services<br>b. Awareness on adolescent sexual reproductive health centres<br>c. Affordability of services<br>d. Health insurance<br>e. Availability of health-related information |
| 3. Need factors influencing the utilization of sexual reproductive health services among female adolescents | a. Perceived need for health services<br>b. View of one's health |

**Social factors.** Social factors refer to the social issues that induce female adolescents to utilize sexual reproductive health services.

Some adolescents described financial difficulty. They exchange sexual favors for financial support. Araba, a seventeen-years-old student recounted:

*Some girls, it is not that they are willing to engage in sexual activities but because of some family problems like financial problems, that's why they have to do that so that the men can pay them for them to be able to take care of their school. I think broken homes too (Araba)*

Harriet further expressed how financial needs have led some of her friends into prostitution:

*My friends, they are into sexual activities. They always go to "ashawo line" (prostitution) and I am their friend but they always advise me that it is not something that is good so I shouldn't worry myself and go into it because now that they are in it, it is difficult to stop. They therefore advise me not to get into it. They sometimes tell me that if I am in big big trouble, I can engage in that. Maybe if I need money, I can go to a man and have sex and use that to solve my problems*

As a result of some of these issues, some participants are exposed to utilization of sexual reproductive health services:

*I took postinor two when I met a boy who said he will take care of me and he forced me to sleep with me and he bought it for me. I tried to arrest him but he approached my parents to take care of me so since JHS, he has been helping to take care of me. After he slept with me, he bought postinor two for me to take. After that, my sister took me to the clinic and they said there is nothing wrong with me, no disease so I was okay. They just educated me about those things (Serwaa)*

Naa explained how an unplanned pregnancy led her to seek abortion services and subsequently resulted in her use of family planning services:

*I got pregnant once and my boyfriend told me I have to abort the baby because he is not ready and me too, I am not ready for that too so, I went to the health center where after taking the pregnancy out, they inserted the family planning for me. They told me to be coming for check up every month, so I go. (Naa)*

**Health beliefs.** The health beliefs held by participants was one of the influencing factors for the utilization of sexual reproductive health services. Some participants expressed that although they experienced some physical changes which are sometimes uncomfortable, they believed it was normal hence did not utilize sexual reproductive health services. Angelina described it as follow:

*And the menses too, it used to be five days but now it is seven or sometimes eight days, yeah. At first it used to be three to five days but now eight and it comes plenty. I thought it is just a normal thing so I just need to use the pads and I am off. I see it as normal so I haven't spoken to anyone about it. Because it doesn't come with any pain, I just have to adjust and I haven't seen anything weird about it so, I have not sought any help. (Angelina)*

Other participants however held health beliefs that indicated a preference to utilization of sexual reproductive health services although they are yet to do so. Jane, a sixteen-years-old student shared that she would want to see a doctor for her menstrual pains although her mother told her it was normal:

*During my menstruation, I get serious pains. I think I have to see a doctor for that. Maybe it is abnormal, maybe it is normal. My mum says it is normal but I don't see it as normal, so I think I need to see a doctor. (Jane)*

Some participants held health beliefs that indicated a preference for some other remedies to the utilization of sexual reproductive health services. Harriet naively shared the following;

*They told me that when the man is having sex with you and he feels like "urinating", that one is the semen so, he has to withdraw. When he withdraws, you won't get pregnant but it is not safe so I don't want to try it. They also said that when you sit on the man, that one too, you will not get pregnant. It is like pouring water into a bottle whiles the bottle is turned with the mouth facing downwards. So, when you are pouring the water into it, it will come out. (Harriet)*

**Knowledge on sexual reproductive health.** This sub theme describes how knowledge of participants on sexual reproductive health influence their predisposition to utilize sexual reproductive health services.

Some participants expressed some level of knowledge on sexual reproductive health;

*Sexual reproductive health to me, it means having sex and protecting yourself from other diseases. Also, you can abstain yourself from sexual intercourse. (Belinda)*

Other participants expressed that they were not sure of what they know about sexual reproductive health. Ama shared:

*Actually, I don't have a fair idea about it but what I know is that it is about your sexual life and how you need to take care of yourself.*

Another participant expressed that she had no knowledge on sexual reproductive health

*I don't really have this thing in it. Like I don't know how to say it, but I think sexual reproductive health is based on how the female reproductive system is. Whether fertile or not. (Juliet)*

**Source of sexual reproductive health information.** Participants shared the various sources of their sexual reproductive health information:

*okay, we learnt about it in social studies and I also heard some of it from friends, not only my age mates but my leaders, pastors, teachers. (Araba)*

Some others also expressed how their source of sexual reproductive health information has been limited to the classroom and expressed a desire for more information:

*All the information I have about sexual reproductive health is what I was taught in school Sometimes, I want education for that. I once went to my mum to talk about those things and even though we spoke a little, I was not really convinced. I searched for information on the internet and I still wasn't so convinced with what I got. After that, I wished I could talk to a healthcare provider, but I can't go alone. (Jane)*

Some participants had to resort to friends as their source of information:

*My parents, they don't have time for me and if I tell them something like that, they will just say I am becoming a bad girl. That's what they say so when I need to talk about these things, I talk to my boyfriend. (Naa)*

**Sexual reproductive health practices.**    Sexual reproductive health practices are actions taken by participants to respond to various sexual reproductive health situations in which they find themselves.

Participants expressed ways in which they deal with some of these situations:

*The white infection (candidiasis) I had was the natural one, yes. They always tell me it is natural, so you have to use water to wash it. My friends, they say it is normal. Like when you tell them that; "the place is itching me ooo", they reply that "as for this one, it is normal". I for instance, it comes more before and during the menses and then it goes after that. I just use water. They say water so I just use water. (Lydia)*

One participant shared how other health related issues tend to receive more attention from her parents whiles little attention is given to issues related to sexual reproductive health. This she said influences her sexual reproductive health practices:

*As for those things that has to do with sexual reproductive health, my mum will say I should get some herbs to drink so I don't go to the hospital for them but because my eyes were paining me and I was shouting and couldn't sleep, they sent me to the hospital yesterday. (Alice)*

Some participants recounted how despite having some sexual reproductive health issues, they are still yet to utilize sexual reproductive health services:

I experience severe pains anytime I menstruate. When it happens, I take paracetamol.

*I mostly get white (Candidiasis). I have bought many medications from the drug store and sometimes, people selling medicines at the roadside, but it is still not going. I haven't been able to go to the hospital to get treatment because I don't like going to the hospital and I feel shy to talk to people about those issues. (Serwaa)*

**Theme 2: Enabling factors influencing the utilization of sexual reproductive health services among female adolescents.**    Enabling factors were described in terms of factors that give female adolescents the ability to utilize sexual reproductive health services. These enabling factors were described in five subcategories; knowledge of adolescent sexual reproductive health services, awareness on adolescent sexual reproductive health centers, affordability of adolescent sexual reproductive health services, health insurance and availability of health-related information.

**Knowledge of adolescent sexual reproductive health services.** Participants shared various levels of knowledge of sexual reproductive health services.

*I know that they give advice and family planning. You know that some of these adolescents, no matter what you tell them, they will not listen so they tell them that if you are in that relationship already and also having sex, then you can do family planning or you get condoms to protect yourself against diseases (Lydia)*

Other participants expressed their lack of knowledge about sexual reproductive health services and indicated that they rather seek help from parents or friends:

*. . .okay, before you try to have sex, you need to seek advice from an elderly person before you go for it. But some people, maybe it is a mistake and it happen, you can go to an elderly person and tell them, maybe she can help you in some way. hmmm, for the pregnancy, I don't know but like for me, I will go to my parents and whatever they will do for me, they will. For the disease too that can happen through sex too, the same thing. I will tell my parents. They will try and get help somewhere for me to go (Belinda)*

One participant expressed a lack of trust in the healthcare system when accessing certain sexual reproductive health services like abortion services:

*If I should get pregnant now, I will go to my friends because I know when I go to see a nurse, she will not help me abort that child so I will go to my friends. I know that as for them, they have been doing it so they will help me (Harriet)*

**Awareness on sexual reproductive health centers.** This sub theme explored how much participants know about the existence of adolescent sexual reproductive health centers where they can access sexual reproductive health services.

*I know of the nurse in my school. She gives us education on how to protect ourselves and abstinence from sex. Sometimes too, when you visit clinics in the community, they try to talk to you about what an adolescent must do like things about sex, the causes and the problems they can give (Lydia)*

For most of the other participants, they had no idea such places existed for adolescents:

*No please, I haven't heard of any such thing. Being in secondary school, if any of such things happen to you, they tell you to go home. For example, if you get "white", they will not tell you to go to a particular hospital or not. I don't think they have such facilities in Tema (Araba)*

Serwaa, expressed her lack of awareness of such centers and also stated her thoughts on the need for awareness:

*I don't know that when you visit the hospital, you can get someone to talk to about these things, just like me, I didn't know when you have sex at the clitoris, it is not actual sex. So, we have to tell others about how these things are including the diseases that are transmitted through sex and how the effects are so that they know (Serwaa)*

**Affordability of services.** Affordability of services was described in terms of how the cost of sexual reproductive health services influenced participant's ability to utilize these services.

One participant described how the unavailability of money prevented her from going for a follow up appointment at the hospital:

*I was having my menses, and I could change my pads like eight times a day, so, I went to a hospital. At the hospital, I was told I had lost a lot of blood so I was given injections. They asked me to go to their bigger facility for a gynaecologist to check some things but when I went home, my mum said she didn't have money, so I didn't go back. (Juliet)*

Serwaa lamented that lack of money sometimes causes her to resort to buying medications from the pharmacy or roadside instead of going to the hospital:

*Because I don't have money sometimes, it prevents me from going to the hospital. Sometimes, I feel like if I have 5 cedis or 10 cedis, I should buy medicine from the drug store or by the roadside for my menstrual pain*

**Health insurance.** Health insurance was discussed to examine the influence of the national health insurance scheme on utilization of sexual reproductive health services among participants.

Some participants expressed their experience generally on the usefulness of the health insurance:

*I have the health insurance and it takes care of some petty petty things when I go to the hospital. The last time I visited the hospital for an asthmatic attack, they only collected 2 cedis and asked me to go and buy some medicine from the pharmacy which they put on me and I felt better. So, I didn't pay much (Harriet)*

Lydia, a sixteen-years-old senior high school student shared her experience of how the health insurance was useful when she visited the clinic for the treatment of candidiasis:

*Yes, I know that the health insurance helps us as adolescents. When my friend visited the clinic for the treatment of the white (Candidiasis), I asked her whether they took money from her and she said she only did the card for 2 cedis but the health insurance covered the cost of treatment*

One participant shared an opposing view indicating that the health insurance is not entirely useful and recounted her experience:

*. . .from my experience and the observations I have made with other people too, even when you have the health insurance, you still have to pay huge amounts of money when you need certain medications. It doesn't show its usefulness (Angelina)*

**Availability of health-related information.** Availability of health-related information was described in terms of how easily accessible information regarding sexual reproductive health is to participants. Most participants expressed that information regarding sexual reproductive health issues is not easily accessible to them as adolescents.

*When I went to the health center, the nurses spoke to me, some were plain. They were two, the other one was feeling shy but it was like the first one, she told me everything like I should keep myself, I should stay with only one guy, I shouldn't go to other men because I think I have done family planning so I can sleep with different men. (Dela)*

For Naa, she only became aware of contraceptive services after she had an unplanned pregnancy and reported to the clinic for abortion services:

*For me, before I got pregnant and went to the clinic, my friends have been talking about things to do to prevent pregnancy, but I didn't know what it was until I got pregnant and went to the health center and they told me*

Another who had a contraceptive implant inserted further shared her grievances on poor access to sexual reproductive health information for adolescents and the difficulty they face in speaking to parents for fear of being labelled as "bad girls":

*I don't think there is enough information on these things. It will be good if you can come and help us because our parents, they don't help us at all. This is because, some of us go through so many things that we cannot even tell our parents. I think it is good we have somewhere that we can go to talk about our problems. Me like this, I can't go and tell my mum or dad that I have inserted family planning. I can't*

**Theme 3: Need factors influencing the utilization of sexual reproductive health services among female adolescents.** Need factors were described in terms of factors that necessitate the use of sexual reproductive health services among female adolescents. Need factors generated two sub themes namely; perceived need for health services and viewpoint of one's health.

**Perceived need for health services.** Perceived need for health services explored participant's perception of how essential sexual reproductive services are.

*I think sexual reproductive health services are important to us because once you damage one part of the reproductive system, it can lead to a whole lot of problems. (Araba)*

*Let's take it for instance, my mum hasn't been to school so if I have some of these problems and I tell her, she will not understand me clearly. If those services are there, those people have been trained so if I talk to them, they will understand. (Juliet)*

Other participants also shared their thoughts on how they think these services could be of help to adolescents:

*I think it is good. This is because, some of us go through so many things that we cannot even tell our parents. I think it is good we have somewhere that we can go to talk about our problems. (Naa)*

**View of one's health.** View of one's health was described in terms of what participants thought about their sexual reproductive health and how these thoughts influence their need for sexual reproductive health services.

Naa, who had done an implant to prevent unwanted pregnancies expressed her fear of the risk of contracting HIV despite having done family planning:

*I have realized that even the family planning will let me get HIV easily because, this one, it only protects against pregnancy. So, if I have sex with someone who has HIV, I can get it easily. I know condom can prevent HIV but my boyfriend will not allow us to use condom. (Naa)*

For Serwaa, despite persistent menstrual pains that sometimes makes it sometimes difficult for her to walk during her menses, she sees it as normal and assumes it will resolve after childbirth.

*There was a time I went to the drug store because I was having menstrual pains and when it comes, I vomit and I can't walk. They told me that it is something normal and that it will go when I give birth. I know I can go to the hospital but hmmm, I feel many people experience it so it is normal. (Serwaa)*

Alice however shared that although she has had recurrent vaginal infections, she does not think it is severe enough for her to visit the hospital for treatment. She thus uses an ointment to treat it;

*Sometimes, I get a certain infection like white. With that one, the toilet we used to go, when I use it, the heat from the toilet makes me to get white. When it comes, I use joy ointment. I don't think it is serious enough for me to go to the hospital (Alice)*

## Discussion

The study revealed poor utilization of sexual reproductive health services among participants, with few participants only utilizing these services after a non-consensual sexual activity or unwanted pregnancy. Several barriers to utilization were also identified. Non-utilization of sexual reproductive health services among adolescents has been identified as a crucial public health concern due to the numerous challenges this poses including unintended pregnancies with its associated outcomes and sexually transmitted infections (STIs) [14].

In exploring factors that predispose participants to the use of SRH services, it was discovered that few of the participants who had utilized some form of sexual reproductive health service were exposed to these services as a result of some unwanated consequence of a sexual encounter with the opposite sex, with most of these sexual relations being used as a means of financial gains. This explains why higher poverty, unsafe neighborhoods and poor social support have been identified to be associated with early age of sexual activity initiation and adolescent pregnancies [15]. All the participants in this study had no source of income but they expressed some financial needs that their parents and guardians are unable to meet, leading to the need to enage in some relationships with the opposite sex in order to meet these needs. Although these factors pushed participants to be sexually active, very few of them utilized SRH services. Participants expressed beliefs that sexual reproductive health is important during adolescence. Adolescent females believe that sexual reproductive health is a priority and that increase access to SHR services is important in improving their SRH outcomes [16]. In as much as participants shared these beliefs, most of the participants were more concerned about measures to prevent unwanted pregnancies and not a comprehensive approach to attain the safest sexual reproductive health. This agrees with other authors who reported that knowledge on SRH services among female adolescents does not neccesarily translate into utilization of services [17]. These further projects that a lot more in addition to information sharing needs to be done to improve utilization of SRH services among adolecsnts. In spite of these beliefs

regarding SRHR, most participants in this study demonstrated low levels of knowledge on sexual reproductive health and services, a phenomenom which negatively impacts participants' utilizataion of SHR services. This is consistent with other studies where they reported that increase in SRH knowledge and increased self-efficacy among adolescents is associated with intentions to reduce sexual risk behaviours as well as increase intention to and use of SRH services [18–20]. Another author identified that the availability of SRH corners for adolescents leads to increased knowledge and use of SRH services among adolescents. All these suggest that a lot more needs to be done to increase knowledge and accessibility to SRH services for adolescents.

Most participants in this study expressed that their source of sexual reproductive health information is limited to what they learn in schools with some participants mentioning friends and some relatives as their source of information. Very few participants mentioned the media and the internet as their source of sexual reproductive health information. This situation leads to most of the participants preferring other means of solving sexual reproductive health issues instead of utilizing SRH services. Similarly, [21] identified that teachers serve as the most common source of sexual reproductive health information among adolescents with parents being the least. This could be due to social norms which seem to view discussions on sexual reproductive health issues as inappropriate. This leads to parents and guardians shying away from having such discussions with adolescents. In spite of the crtical role teachers seem to play in SRH information sharing among adolescents, Most of the study participants lamented about the poor availability of sexual reproductive health information, leading to their reliance on peers for such information. Consequently, participants prefer to seek solutions to SRH problems from their peers compared to utilizing SRH services. Participants expressed that availability of adequate information on sexual reproductive health services will go a long way to influence their sexual reproductive health decision making and subsequent use of sexual reproductive health services.

Almost all the participants in the current study shared varied factors that influence their sexual reproductive health practices. Some of these factors include the degree of severity they attach to their sexual reproductive health problems, the amount of attention the reproductive health problem they are having receive from their parents as well as the degree of urgency to seek health care posed by the problem. Participants further shared how other health related problems tend to receive more attention from parents than sexual reproductive health problems. This seems to highlight the need for parents and guardians to be educated on adolescent sexual reproductive health needs. This result agrees with a report that multiple levels of social influence impact adolescent sexual reproductive health decision-making and behaviors. These factors include interpersonal, community and macro-social level influences that impact adolescent girl's SRH decision making and behaviors [22].

In the current study, most of the participants had little to no awareness on sexual reproductive health centres. The few participants who had little awareness mentioned school based clinics as their source of sexual reproductive health services. This little to no awareness resulted in most participants seeking help outside SRH facilities. Similarly, Othman, Kong [23] revealed that in Malaysia, only one out of ten adolescents was aware of the availability of sexual reproductive services. Findings from another study in the Lao PDR further suggested that lack of awareness of sexual reproductive health services among adolescents is a source of cognitive accessibility barrier to the utilization of these services [24]. These suggest the need for extensive awareness creation on SRH services among adolescents. Although participants in the current study seem to rely heavily on school based clinics for sexual reproductive health services, existing evidence on the effectiveness of school based SRH inervention programmes in improving SRH outcomes of adolescents showed that these programmes alone are not completely

effective [25]. Most of the paricipants further expressed their desire for improvements in awareness creation on sexual reproductive health services in order to make it easy for them to access these services when the need arises.

Another interesting finding from the study was that in terms of affordability of sexual reproductive health services, most of the participants lamented that their inability to afford some sexual reproductive health services contributed to non utilization of the services. Two of the participants in the current study who had utilized abortion services stated that they had to pay out of pocket. Existing literature suggests that a strong relationship exist between access to low cost services and utilization of sexual reproductive health services [26]. In spite of this, the essential package of health care in many countries excludes critical sexual reproductive health services such as safe abortion of which Ghana is no exception. In a similar vein, [27] reported the impact of out of pocket payment for health services among female adolescents, leading to impoverishment and consequently a higher likelihood of not utilizing health services. Inadequate international and domestic public funding of sexual reproductive health services contributes to a continual burden of self paying expenditure and inequities in access to sexual reproductive health services [28]. Most participants further added that due to their inability to afford some sexual reproductive health services, they either could not honor a follow up visit or resorted to buying medications from a pharmacy or by the roadside as a remedy for their sexual reproductive health problems.

## Conclusion

The study explored utilization of sexual reproductive health services among female adolescents in the Tema metropolis in Ghana. Findings from the study revealed low utilization of SRH services among female adolescents in the Metropolis. Few participants who had utilized any sexual reproductive service only did so after either a non-consensual unprotected sexual activity or an unwanted pregnancy. Barriers to utilization of sexual reproductive health services among these adolescents include lack of awareness on sexual reproductive health services, unreliable sources of SRH information, underestimation of the severity of sexual reproductive health problems faced, unmet expectations and poor experiences with service providers. The study recommends the need to increase the number of adolescent health corners, increase awareness about SRH services among adolescents, improve parent-child SRH communication and provide adequate training for healthcare providers to improve provider attitude towards adolescent SRH service delivery in order to increase utilization of sexual reproductive health services among female adolescents in the metropolis.

### Strengths of the study

One strength of the current study is that the sexual reproductive health service concerns of a vulnerable group (female adolescents) have been explored and narrated. These narratives are essential because the review of the literature suggests that this is a grey area in Ghana.

### Limitations of the study

The sensitive nature of some questions asked to explore aspects of participants' sexual reproductive health may have caused some participants to withhold some sensitive but important information. However, measures such as adequate rapport building with participants prior to interviews, conducting interviews in a friendly manner and use of appropriate probes where necessary were employed to curb the situation.

## Supporting information

**S1 File.**
(ZIP)

## Acknowledgments

The authors deeply acknowledge the participants who took part in the study, the research and adolescent health department of the Tema Metropolitan Health Directorate, and the entire staff of the directorate.

## Author Contributions

**Conceptualization:** Innes Agbenu, Josephine Kyei, Florence Naab.

**Formal analysis:** Innes Agbenu.

**Methodology:** Innes Agbenu, Josephine Kyei.

**Supervision:** Josephine Kyei, Florence Naab.

**Writing – original draft:** Innes Agbenu, Josephine Kyei.

**Writing – review & editing:** Innes Agbenu, Josephine Kyei, Florence Naab.

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
