## [Decision Letter · Decision Letter 0]

6 Feb 2023

PONE-D-22-32401Adolescent sexual reproductive health service utilization concerns; a qualitative enquiry.PLOS ONE

Dear Prof. Naab

Thank you for submitting your manuscript to PLOS ONE. After careful consideration, we feel that it has merit but does not fully meet PLOS ONE’s publication criteria as it currently stands. Therefore, we invite you to submit a revised version of the manuscript that addresses the points raised during the review process.

Please, ensure that all comments made by the two reviewers are adequately addressed and your revised manuscript is thoroughly proofread before resubmission. 

We look forward to receiving your revised manuscript.

Kind regards,

Gilbert Abotisem Abiiro, PhD

Academic Editor

PLOS ONE

Journal Requirements:

2. a) Please clarify whether you have written consent for publication for use of the participants names in the study. For further information please refer to our policy on informed consent for publication https://journals.plos.org/plosone/s/human-subjects-research#loc-Patient-Privacy-and-Informed-Consent-for-Publication;
" ext-link-type="uri" xlink:type="simple">https://journals.plos.org/plosone/s/file?id=8ce6/plos-consent-form-english.pdf"

b) During your revisions, please confirm whether the wording in the title is correct and update it in the manuscript file and online submission information if needed. Specifically, the submitted title of the manuscript is "The Editor, PLOS ONE JOURNAL", please update this to the correct manuscript title.

c) You indicated that you had ethical approval for your study. In your Methods section, please ensure you have also stated whether you obtained consent from parents or guardians of the minors included in the study or whether the research ethics committee or IRB specifically waived the need for their consent.

Reviewers' comments:

Reviewer's Responses to Questions

**Comments to the Author**

1. Is the manuscript technically sound, and do the data support the conclusions?

Reviewer #1: Yes

Reviewer #2: Partly

2. Has the statistical analysis been performed appropriately and rigorously? 

Reviewer #1: N/A

Reviewer #2: N/A

3. Have the authors made all data underlying the findings in their manuscript fully available?

Reviewer #1: Yes

Reviewer #2: Yes

4. Is the manuscript presented in an intelligible fashion and written in standard English?

Reviewer #1: Yes

Reviewer #2: Yes

5. Review Comments to the Author

Reviewer #1: Feedback on ‘Adolescent sexual reproductive health service utilization concerns: A qualitative enquiry

The authors brought out an important issue but the write-up was generally not well written.

Title

Full title: There was no full title and the short title provided did not indicate where the study was conducted and who the study population are. Also, the title is not well written and authors should revise it based on their main objective.

Suggested title: Perceived factors influencing utilization of adolescent sexual and reproductive health service among female adolescents in the Tema Metropolis: A qualitative enquiry.

Abstract

Abstract: Check the grammar under the methods segment. Which metropolis in southern Ghana was the study conducted? Why did they use a semi-structured interview guide to conduct in-depth interviews? There is a difference between content analysis and thematic analysis. The authors should let us know the type of analysis that was done.

The results should contain only the main findings, so the 1st sentence is not required. Factors/barriers to using health services should be properly stated.

Conclusions-The authors should briefly state the key findings and their implications.

Introduction

Please, the authors should edit sentence 57.

Line 74, what initiatives have been undertaken to improve adolescents’ health? What are the effects of these initiatives? Line 83, the grammatical error should be corrected.

84 which metropolis was the study conducted in? Line 84-86, the authors should provide current information about the metropolis. Thus, the introduction should be strengthened with up-to-date statistics since that is readily available. Also, there are several related studies in Ghana but the authors did not include their findings and gaps in previous studies that make the current study relevant.

Methods

The methods are not sufficient enough to replicate this study and for that matter, the authors must expand on it. Which metropolis was the study done in? Where is it located? How many health staff, youth corners etc?

Study setting: This is not well articulated. Sentence 108 should be edited. The Ghana Health Service annual report from the metropolis could be used to expand on why the study is being conducted.

Ethical considerations: A separate heading should be created to discuss ethical issues. Ethical issues should not be blended with procedures for data collection.

Procedures for data collection: How was the in-depth interview guide developed? Was it from literature? Was the tool pre-tested? What language were the interviews conducted in? In all, how many interviews were conducted?

Data Analysis: In the methods, authors talked about content analysis that was performed but in the abstract, they talked about content thematic analysis. They should resolve the discrepancies. They should elaborate on how data was analyzed. Please, edit sentences 131 and 132. What were these codes? Describe the three themes and 12 sub-themes.

Results

This was not well written. Authors should describe the demographic features of the respondents before they present the actual results based on their objectives.

144-149- The themes should be described under the analysis.

Check sentence 152.

Avoid using the names of respondents as you present the results.

There is no full sentence for sentence 188.

Quotes from participants had no full stops.

Authors should avoid describing terms under the results and tell us exactly what the respondents said about those issues.

274-290; the quote is too long. The main issues said by the respondent should be projected.

Discussion

The main findings should be extensively discussed and the implications of the study must stand out.

Conclusions

The implications of the study were not projected.

Study strengths and limitations

These were not projected.

References

The citations are not in line with the requirements of the journal;Vancouver.

Thank you.

Reviewer #2: Reviewer Comments:

Adolescent sexual reproductive health service utilization concerns: A qualitative

Enquiry

Abstract:

The abstract is well-written but there are a couple of typos/grammatical errors that appear to derail is flow.

Major issues

Introduction

Reviewer comment 1:

The introduction of the paper is quite OK, and the knowledge gap appears to be well established. However, a major weakness in this section is that given the specifics of the study, the authors needed to situate the work in a clear theoretical/conceptual framework.

Methods

Reviewer comment 2:

The research design is appropriate for the study. However, the authors mentioned in the abstract in line 38 that a purposive sampling technique was used to select 12 adolescents for interviews. The rationale/justification for the use of purposive sampling and the sample size of 12 participants have not been provided. For example, what informed the choice of 12 participants—why not 20, 8, 40, etc but 12?

Results:

Theme 1: Predisposing factors influencing the utilization of sexual reproductive health

services among female adolescents

Reviewer comment 3:

The theme above clearly sets out to explore how various predisposing factors influenced the use of sexual reproductive health services among female adolescents. However, reading through the theme specifically from lines 157-165, it appears the authors have not sufficiently addressed the research question. In particular, in lines 169-181, the authors rather seem to explain the reasons that compel the adolescent girls to engage in sexual activities—for example, financial difficulty (poverty) and not the factors that predisposed them to use SRH services as captured clearly in the study’s aim in line 35 of the paper. The quotes in lines 171-181 do not reflect the preamble/explanation in lines 166-168 under the social factors. The same concerned is observed from lines 184-210.

Reviewer comment 4:

Similarly, from lines 211-224, the authors appear interested in establishing the awareness levels of SRH among the adolescents rather than how their knowledge of SRH predisposed them to using the SRH service as the quotes illustrate.

Reviewer comment 5:

Again, lines 225-290 appears to explore the adolescents’ source of knowledge about SRH and not how their knowledge about SRH influences their use of SRH services.

Theme 2: Enabling factors influencing the utilization of sexual reproductive health services among female adolescents

Reviewer comment 6:

While the explanation of the major theme provided by the authors in lines 307-311 is satisfactory, the explanation of the sub-theme “Knowledge of adolescent sexual reproductive health services” lines 312-319 do not support the major theme. Naturally, it would have been interesting to see how the knowledge of the adolescents influenced the use of SRH services or otherwise. The quote in lines 316-319 hardly contributes to the major theme.

Reviewer comment 7:

The sub-theme “Awareness on sexual reproductive health centers” attempts to determine whether or not adolescents were aware of the existence of reproductive health centres rather than the use of SRH services. The point here is that “awareness” and “utilisation” are distinct variables because the adolescents may be aware of the existence of the reproductive health centres but fail to use them. The focus should have been to establish how the awareness influenced the use of the SRH. The quotes that follow in lines 338-359 do very little to address the major concern regarding the use of SRH services.

Reviewer comment 8:

While the sub-themes “Affordability of services” and “Health insurance” appropriately contribute to the major theme above, the quotes in lines 366-377 and lines 379-393 that seem to support these sub-themes do not strongly reflect the major theme. It would have been interesting if the authors provided quotes that directly speak to these sub-themes, they are quite straight forward and easy to comprehend. In addition, the selected quotes need to address the topic directly “use of SRH” services but they (quotes) are generic because in lines 383-384, the authors generally explain the advantages of possessing a valid health insurance as it helps to take care of “petty petty things” and taking care of asthmatic attacks. The fact here is that the authors focus more on the usefulness of health insurance but do not go further to establish how the “usefulness” of health insurance enhances their use of SRH services.

Theme 3: Need factors influencing the utilization of sexual reproductive health services among female adolescents

Reviewer comment 9:

The authors presented two sub-themes under the major theme above—"Perceived need for health services” line 436 and “View of one’s health” line 458. These sub-themes are actually the same because one’s view of health simply describes the person’s perception about his/her state of health—whether or not people perceive their health problems to be of sufficient importance or magnitude to seek professional health. In fact, the authors’ explanation of the two sub-themes and the narratives/quotes that are selected to support their claims speak to the same issue about the “perceived need” for SRH services for adolescent girls as contained in lines 436-483. In the literature, the two types of need factors that have been identified include—perceived need and evaluated need (see Andersen, 1995; Alatinga et al., 2021).

Discussion:

Reviewer comment 10:

The major concern in the discussion section relates to the fact that the discussions do not adequately reflect the results. For example, in line 486, the authors categorically stated that “…. revealed poor utilization of sexual reproductive health services among participants” but this claim does not clearly stand out in the results. Again, in line 487, the authors state that “…. Several barriers to utilisation were also identified” yet in the results, the authors only highlighted “financial difficulties” as the major reason for which adolescent do not use SRH services.

Reviewer comment 11:

Another major issue in the discussion section is that, the discussions are not anchored on the 3 themes that have been identified in the results. In fact, the discussions have largely been done independent of these themes. For example, in lines 492-497, the authors aver to issues of love, intimacy and belonging, high levels of poverty, etc which are not directly reflected in the results. For example, “In exploring factors that predispose participants to the use of SRH services, it was discovered that in addition to love, intimacy and belonging, most of the participants who were involved in sexual relationships did so for financial gains. This explains why higher poverty, unsafe neighborhoods and poor social support have been identified to be associated with early age of sexual activity initiation and adolescent pregnancies (Decker et al., 2018). All the participants in this study had no source of income but they expressed some financial needs that their parents and guardians are unable to meet,…..”. These explanations in my opinion, are far remote. Plus, granted that adolescents engaged in sexual relationships for financial gains, the authors do not make an impelling case as to how these financial gains enhanced their use of the SRH services or otherwise. For these reasons, there appears to be a disconnect between the themes/results and the discussions.

Reviewer comment 12:

The disconnection between the results and discussions is even more pronounced from lines 529-545, where the authors aver to issues of comprehensive sexuality education. In my opinion, this large disconnect between the results and discussions steps from the factor that the paper is not theoretically/conceptually robust. Because while the paper is anchored on the so-called behavioural model and access to medical care—predisposing factors, enabling/inhibiting and need based factors, the relevant literature that discusses this model is not even cited in the entire paper (See for example, Andersen, 1995; Andersen Newman, 2005; Alatinga et al., 2021). These three references clearly and logically present the three themes of the current paper, and so its theoretical weaknesses result from the fact that the right literature was not consulted to ground it.

Conclusion

Reviewer comment 13:

The disconnect between the results and the discussions follow through to the conclusion. The conclusion provided in this paper do not adequate emanate from the results—largely, the results do not support the conclusion, and the recommendations provided are very generic and not action-oriented. Because of the disconnect between the results, discussions and conclusion, in my opinion, the paper in its current state adds very little to knowledge in the subject matter, especially in Ghana.

Minor issues

Suggested title:

Female adolescent sexual reproductive health service utilization concerns: A qualitative Enquiry in the Tema Metropolis in Ghana

Age range of target population

In the methods section, the authors report that female adolescents aged 14-19 were selected for the study but in the results section, the authors report that adolescents aged 14-18 (line 151) were selected. A reconciliation is needed

Typos and Grammatical Errors

The paper will benefit from a general language edit because there are grammatical errors and typos. See lines; 57-58, line 121, lines 140-140, line 185, line 286, line 406, etc

References

The in-text citation in line 66 is not correct. It’s mixed with the journal title.

The reference list needs to be carefully formatted.

6. PLOS authors have the option to publish the peer review history of their article (what does this mean?). If published, this will include your full peer review and any attached files.

Reviewer #1: No

Reviewer #2: **Yes: **Kennedy A. Alatinga

---

## [Author Response · Author response to Decision Letter 0]

16 Mar 2023

Response to reviewers has been attached as a separate document.

---

## [Decision Letter · Decision Letter 1]

26 Apr 2023

PONE-D-22-32401R1Female adolescent sexual reproductive health service utilization concerns: a qualitative enquiry in the Tema Metropolis in Ghana.PLOS ONE

Dear Prof Florence Naab,

Thank you for submitting your manuscript to PLOS ONE. After careful consideration, we feel that it has merit but does not fully meet PLOS ONE’s publication criteria as it currently stands. Therefore, we invite you to submit a revised version of the manuscript that addresses the points raised during the review process.

We look forward to receiving your revised manuscript.

Kind regards,

Gilbert Abotisem Abiiro, PhD

Academic Editor

PLOS ONE

Journal Requirements:

Reviewers' comments:

Reviewer's Responses to Questions

**Comments to the Author**

1. If the authors have adequately addressed your comments raised in a previous round of review and you feel that this manuscript is now acceptable for publication, you may indicate that here to bypass the “Comments to the Author” section, enter your conflict of interest statement in the “Confidential to Editor” section, and submit your "Accept" recommendation.

Reviewer #1: (No Response)

Reviewer #2: All comments have been addressed

2. Is the manuscript technically sound, and do the data support the conclusions?

Reviewer #1: Partly

Reviewer #2: Yes

3. Has the statistical analysis been performed appropriately and rigorously? 

Reviewer #1: No

Reviewer #2: N/A

4. Have the authors made all data underlying the findings in their manuscript fully available?

Reviewer #1: No

Reviewer #2: Yes

5. Is the manuscript presented in an intelligible fashion and written in standard English?

Reviewer #1: No

Reviewer #2: Yes

6. Review Comments to the Author

Reviewer #1: Female adolescent sexual and reproductive health service utilization concerns: a qualitative enquiry in the Tema Metropolis in Ghana.

Generally, authors have improved upon the write-up. However, there are still some issues that require redress. The comments below should be addressed to strengthened the quality of the article. Sexual Reproductive Health (SRH) should always be defined as sexual and reproductive health.

ABSTRACT

Background Information

Authors could add “…and yet evidence on utilization of SRH services among female adolescents is sub-optimal”.

Methods

Authors should check the difference between semi-structured interviews and unstructured interviews. Also, there is a difference between thematic analysis and thematic content analysis.

Results: Since this is a qualitative study, authors should indicate “that perceived utilization of SRH services was low”.

Conclusion

The last sentence of the conclusion has no full stop. Perceived factors of utilization of SRH should be captured under the conclusion.

ARTICLE

Introduction

Sexual reproductive health services (SRHS) should be written as Sexual and Reproductive Health Service. Adolescent Sexual Reproductive Health should also be captured as Adolescent Sexual and Reproductive Health.

Worldwide adolescents face numerous challenges…… Authors should also add the challenges faced by adolescents in the African context including Sub Saharan Africa before giving us the picture in Ghana.

The study focuses on utilization of sexual and reproductive health services among female adolescents and perceived factors of utilization. But under the introduction, I have not seen any previous related evidence around this area. There is so much published evidence around it but authors did not include any of it. For instance, what related works have been done elsewhere including Ghana about the topic? What are the gaps in these related works that the study seeks to address?

The Anderson and Newman behavioral health services utilization model was used…. Sentences about the model were not referenced at all. All factual statements made by authors should have citations. Also, the sentence about the model should be put at an appropriate location. Where it is currently does not make the reading of the article interesting.

Study setting

There are factual statements without citations. Authors should fix them.

Data Collection Procedures

The data collection tool should be provided.

Interview guides were developed based on the constructs of the conceptual model used, reviewed literature… Provide citation. But the conceptual model being referred to is rather a theoretical model.

Findings

Authors should put the explanation below under data collection procedures. The results aspect of the article is only meant for presentation of results;

‘Predisposing factors refer to factors that influence either positively or negatively utilization of services. Enabling factors are factors that make utilization of services easier. Need factors are situations that propel an individual to service utilization’.

Demographic characteristics

‘Nine were senior high school students and were junior high school graduates’. Authors should show the table for the demographic features.

There should be a preamble or an appropriate topic sentence or heading before introducing the Table 1.

Authors should critically look at the themes and sub themes again. Those that are not really perceived factors of utilization of SRH should be removed. Also, I have not seen a theme on perceived utilization of SRH. Under the results, authors said it was low but I have not actually seen detail findings on it.

Discussion

The discussion segment should be focused on main findings on utilization of SRH services and the perceived factors. But other issues not related to the study objectives were discussed. Please, correct ‘unwanated consequence’.

Conclusions

… The study revealed low utilization of SRH services...Add the full stop to the last sentence.

Strengths and limitations of the study

This should be strengthened because what is currently captured is inadequate.

References

Some citations are Vancouver whilst others are not. The journal uses the Vancouver referencing style and so authors should revise citations based on this recommended style. The reference list should be reorganized.

Reviewer #2: The comments have been satisfactorily addressed. However, the manuscript will still benefit from a language edit because there are few typos/grammar errors.

1. Under the research design, line 4, page 4, allow should be "allowed"

2. Under the research setting, also line 4 page 4, has should be "had"

3. Under procedure for data collection, line 8 page 5, 12th participants should be 12th participant (singular not plural).

4. Under Demographic characteristics of respondents, line 2 page 6 "three" is missing in the sentence "Nine were senior high school students and THREE were....."

5. Under social factors in page 8 line 3, the sentence "Some adolescents described financial difficulty. They exchange sexual favors for financial" is not complete.

6. The issue of the resistance of the introduction of comprehensive sexuality education in Ghana on page 17 appears very remote to the objectives of the study. Besides, it was not even mentioned in the background/conceptual discussion, hence it appears misplaced in the discussion. I would have deleted.

7. The authors should read the manuscript again and identify the key strengths because just stating that the conceptual framework is its strength, is not robust enough.

7. PLOS authors have the option to publish the peer review history of their article (what does this mean?). If published, this will include your full peer review and any attached files.

Reviewer #1: No

Reviewer #2: **Yes: **Kennedy A. Alatinga

---

## [Editor Report · Decision Letter 2]

25 May 2023

PONE-D-22-32401R2Female adolescent sexual reproductive health service utilization concerns: a qualitative enquiry in the Tema Metropolis of Ghana.PLOS ONE

Dear Prof Naab,

Thank you for submitting your manuscript to PLOS ONE. After careful consideration, we feel that it has merit but does not fully meet PLOS ONE’s publication criteria as it currently stands. Therefore, we invite you to submit a revised version of the manuscript that addresses the points raised during the review process. I have observed that you did not respond to the second round of comments from the reviewers. Please provide a response to each of the comments by the reviewer in your response to reviewers document. 

We look forward to receiving your revised manuscript.

Kind regards,

Gilbert Abotisem Abiiro, PhD

Academic Editor

PLOS ONE

---

## [Editor Report · Decision Letter 3]

14 Sep 2023

Female adolescent sexual reproductive health service utilization concerns: a qualitative enquiry in the Tema Metropolis of Ghana.

PONE-D-22-32401R3

Dear Prof. Florence Naab,

We’re pleased to inform you that your manuscript has been judged scientifically suitable for publication and will be formally accepted for publication once it meets all outstanding technical requirements.

Kind regards,

Gilbert Abotisem Abiiro, PhD

Academic Editor

PLOS ONE
---

## [Editor Report · Acceptance letter]

15 Feb 2024

PONE-D-22-32401R3 

PLOS ONE

Dear Dr. Naab, 

I'm pleased to inform you that your manuscript has been deemed suitable for publication in PLOS ONE. Congratulations! Your manuscript is now being handed over to our production team.

Kind regards, 

on behalf of

Dr. Gilbert Abotisem Abiiro 

Academic Editor

PLOS ONE